# Influence of the Gas Bubble Size Distribution on the Ladle Stirring Process

**Mengkun Li** [1,*,†] [ID] **and Lintao Zhang** [2,*,†]

1 School of Management, Capital Normal University, Beijing 100089, China
2 Materials Advanced Characterisation Centre (MACH1), College of Engineering, Swansea University, Bay Campus, Fabian Way, Swansea SA1 8EN, UK
* Correspondence: limengkun@cnu.edu.cn (M.L.); L.Zhang@swansea.ac.uk (L.Z.)
† These authors contributed equally to this work.

**Abstract:** This work aims at figuring out the influence of gas bubble size distribution on the ladle stirring process. The work is conducted through three-dimensional (3D) numerical simulation based on the finite volume method. Mesh sensitivity test and the cross-validation are performed to ensure the results are mesh independent and the numerical set-up is correct. Two distributions, uniform and Log-normal function, are investigated under different gas flow rates and number of porous plugs. The results indicate that the results, e.g., the axial velocity and the area of the slag eye, have little difference for low flow rate. The difference becomes dominant whilst the flow rate is increasing, such as 600 NL/min. The Log-normal function bubble size distribution gives a larger axial velocity and a smaller slag eye area compared to the uniform bubble size distribution. This work indicated that, at a higher flow rate, the Log-normal function is a better choice to predict the melt behavior and the slag open eye in the ladle refining process if the bubble interaction is not considered.

**Keywords:** secondary metallurgy; numerical simulation; ladle refining; ladle bottom stirring; bubble diameter

## 1. Introduction

Steel refining in the ladle is an important process in the secondary metallurgy. It usually happens before the liquid steel is poured into the tundish and it may last from half an hour to several hours [1]. The gas, Argon for instance, is usually used to stir the liquid steel with the aims such as uniform bulk melt temperature and removing the non-metallic inclusions. The Argon is usually injected through the porous plugs installed in the ladle bottom and the idea was proposed by Spire [2]. During the gas stirring process, the heat and mass transfer are very important [3] and a lot of factors could affect this process, the number of the porous plugs [4], the plug location [5], the gas flow rate [6], and the slag properties (e.g., slag height) [7], for instance. For the cases with dual plugs, the influence of the non-uniform gas flow rate for each plug has also been discussed [8]. The extensive work has been conducted to set up the relationship between different parameters, such as the slag eye area and the mixing time with the gas flow rate [9–14].

During the process, the gas bubble behavior is vital because it is the main factor to trigger the stirring process. Therefore, the bubble behavior attracted a lot attention. Xie et al. conducted the experiment, by adopting Wood alloy and nitrogen to simulate the liquid steel and the argon, to investigate the bubble behavior in the process [15]. The results showed that bubble size distribution obeyed a Log-normal function. Liu et al. conducted the numerical simulations and the results indicated that the final bubble distribution also obeyed the Log-normal function and it is independent from the initial/injected bubble size [16]. This bubble size distribution is due to the bubbles' interactions, such as

the breakup and the coalescence, which is a highly complex process and not yet fully understood. In addition, some work did not take the bubble interactions into account. Schalk et al. [17] developed a three-dimensional model to simulate the gas stirring process with higher computational efficiencies. Singh et al. [18] studied the shear stress in the vicinity of the ladle walls by employing different meshes with the aim to obtain more precise values. Aoki et al. [19] studied the mixing behavior in detail through both the experiment and theoretical method. The simulations without bubble interactions can improve the commutating efficiency significantly. However, this triggers several questions:

1. How does the initial injected bubble size distribution, e.g., uniform v.s. Log-normal function, affect the final result whilst no bubbles' interactions are being taken into account?
2. If there is a difference, how does the difference vary with other parameters, such as flow rate and number of the porous plugs?

The current research aims at tackling the above questions. To the best of the authors' knowledge, there is no such research that has been done before. The outline of the present paper is as follows: In Section 2, the geometry and numerical set-up are discussed. This consists of the dimension of the ladle, the governing equations, the mesh sensitivity test, the cross-validation, the initial and boundary conditions, and the numerical procedures. Section 3 discusses the overview of the argon injection and stirring processes. The bubble diameter possibility and the bubble distribution and the slag eye information are compared in detail in Section 4. The slag eye information is discussed in Section 5. Finally, the main conclusions are summarized in Section 6.

## 2. Geometry and Numerical Set-Up

### 2.1. Geometry

Figure 1 showed the sketch of the adopted ladle geometry. The unit of the dimension is in millimeters.

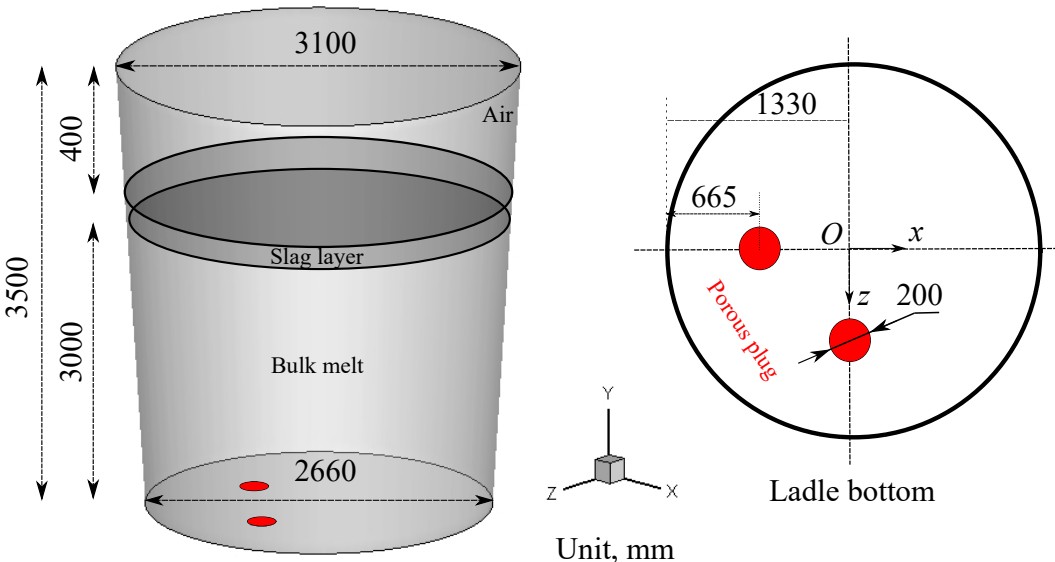

**Figure 1.** Sketch of the ladle (not scaled).

The origin (*O*) is located at the middle of the ladle bottom. *x* and *z* are the directions pointing to the ladle wall. The *y*-axis is the direction pointing to the ladle top surface. The center of the porous plug is located at the middle point of the radius of the ladle bottom. The two plugs have an angle of 90° (separation angle). This layout proved to be efficient [13]. Table 1 shows the ladle dimensions and the material properties for the air, slag, liquid steel, and argon used in the current work.

**Table 1.** The dimensions of ladle and other parameters adopted in the simulation.

| | | Unit | Value |
|---|---|---|---|
| | Ladle top diameter | mm | 3100 |
| Ladle | Ladle bottom diameter | mm | 2660 |
| | Ladle height | mm | 3500 |
| | Air layer thickness | mm | 400 |
| Air | Air viscosity | kg/(m·s) | $1.7894 \times 10^{-5}$ |
| | Air density | kg/m$^3$ | 1.225 |
| | Slag layer thickness | mm | 100 |
| Slag | Slag viscosity | kg/(m·s) | 0.006 |
| | Slag density | kg/m$^3$ | 3500 |
| | Liquid steel layer height | mm | 3000 |
| Liquid steel | Liquid steel viscosity | kg/(m·s) | 0.03 |
| | Liquid steel density | kg/m$^3$ | 7020 |
| Argon | Argon viscosity | kg/(m·s) | $2.125 \times 10^{-5}$ |
| | Argon density | kg/m$^3$ | 1.6228 |

*2.2. Numerical Set-Up*

The following assumptions were made:

1.  the problem was transient and isothermal with a constant temperature of 1873 K;
2.  the molten steel was homogeneous, viscous, and incompressible fluid;
3.  the physical properties are isotropic and constant;
4.  the interaction between bubbles, e.g., breakup and coalescence, were ignored;
5.  the turbulence kinetic energy induced by the bubble was ignored;
6.  only the liquid slag layer was considered.

2.2.1. Governing Equations

The interfaces of the liquid steel, the slag, and the air were captured through the volume of fluid (VoF) method. The VoF method has been proved as an effective method to capture the interface behavior between difference phases. The volume fraction equation for each phase is solved:

$$\frac{1}{\rho_i}\left[\frac{\partial}{\partial t}(\alpha_i \rho_i) + \nabla \cdot (\alpha_i \rho_i \mathbf{u}_i)\right] = 0, \tag{1}$$

where $\rho_i$, $\alpha_i$, and $\mathbf{u}_i$ stand for the '$i$' phase density, volume fraction, and velocity, respectively. The volume fraction in a given cell is constrained by the following equation:

$$\sum \alpha_i = 1. \tag{2}$$

The continuity equation and momentum equation coupled with mixing density $\rho$ and mixing viscosity $\mu$ are solved throughout the domain, and the velocity field is shared for all phase by weighting their volume fraction. The continuity equation and momentum equation can be described as:

$$\frac{\partial \rho}{\partial t} + \nabla \cdot (\rho \mathbf{u}) = 0, \tag{3}$$

$$\frac{\partial(\rho \mathbf{u})}{\partial t} + \nabla \cdot (\rho \mathbf{u}\mathbf{u}) = -\nabla p + \nabla \cdot [(\mu + \mu_t)(\nabla \mathbf{u} + \nabla \mathbf{u}^T)] + \rho \mathbf{g} + \mathbf{f}_\sigma + \mathbf{F}_b, \tag{4}$$

$$\rho = \sum \alpha_i \rho_i, \tag{5}$$

$$\mu = \sum \alpha_i \mu_i, \tag{6}$$

where $\rho$, $t$, $\mathbf{u}$, $p$, $\mu$, $\mu_t$, $\mathbf{g}$, $\mathbf{f}_\sigma$, $\mathbf{F}_b$, and $\mu_i$ stand for density, time, velocity, pressure, viscosity, turbulent viscosity, gravity acceleration, the momentum source resulting from surface tension, the interaction

force between bubble, and liquid phase, which can be obtained from the force endured by bubbles with equal value and opposite direction and the viscosity of the '*i*' phase, respectively.

The standard two-equation $k - \epsilon$ model with a scalable wall function is applied to describe the turbulent behavior of the flow field. The turbulent kinetic energy '*k*' and the dissipation rate of turbulent kinetic energy $\epsilon$ can be formulated as:

$$\rho \frac{\partial k}{\partial_t} + \nabla \cdot (\rho \mathbf{u} k) = \nabla [(\mu + \frac{\mu_t}{\sigma_k}) \nabla k] + G_k - \rho \epsilon, \tag{7}$$

$$\rho \frac{\partial \epsilon}{\partial_t} + \nabla \cdot (\rho \mathbf{u} \epsilon) = \nabla [(\mu + \frac{\mu_t}{\sigma_\epsilon}) \nabla \epsilon] + \frac{\epsilon}{k}(C_1 G_k - C_2 \rho \epsilon), \tag{8}$$

where $\sigma_k$, $\sigma_\epsilon$, $C_1$, and $C_2$ have constant values 1.0, 1.3, 1.44, and 1.92, respectively. The turbulent viscosity $\mu_t$ and the turbulent kinetic energy $G_k$ can be expressed by:

$$\mu_t = \rho c_\mu \frac{k^2}{\epsilon}, \tag{9}$$

$$G_k = \mu_t \frac{\partial u_j}{\partial x_i}, \tag{10}$$

where $c_\mu$ is a constant with a value of 0.09.

The movement of bubble is decided by Newton's second law of motion and tracked by the Lagrangian approach. The motion trajectory of bubble can be calculated by the integration of time yield and its velocity.

The trajectory equation and the Lagrangian form of Newton's second law of motion can be expressed as:

$$\mathbf{x}_b = \int \mathbf{u}_b dt, \tag{11}$$

$$\frac{d\mathbf{u}_b}{dt} = \mathbf{F}_G + \mathbf{F}_B + \mathbf{F}_D + \mathbf{F}_{VM} + \mathbf{F}_P + \mathbf{F}_L, \tag{12}$$

where $\mathbf{x}_b$, $\mathbf{u}_b$, $\mathbf{F}_G$, $\mathbf{F}_B$, $\mathbf{F}_D$, $\mathbf{F}_{VM}$, $\mathbf{F}_P$, and $\mathbf{F}_L$ are the motion trajectory in an integration time, the bubble velocity, the gravity, the buoyancy force, the drag force, the virtual mass force, the pressure gradient force, and the lift force, respectively. The combined effect of the gravity and buoyancy force on bubbles can be expressed as:

$$\mathbf{F}_G + \mathbf{F}_B = \frac{\rho_b - \rho}{\rho_b} \mathbf{g}, \tag{13}$$

where $\rho_b$ is the bubble density. The drag force between bubbles and fluid depends on bubble Reynolds number $Re_b$, fluid viscosity, bubble density, and slip velocity between bubbles and fluid. The drag force can be written as:

$$\mathbf{F}_D = \frac{18 \mu C_d Re_b}{24 d_b^2 \rho_b} (\mathbf{u} - \mathbf{u}_b), \tag{14}$$

$$Re_b = \frac{\rho d_b |\mathbf{u} - \mathbf{u}_b|}{\mu}, \tag{15}$$

where $d_b$ is the bubble diameter and $C_D$ is the drag coefficient. The Kuo and Wallis model is adopted with considering bubble shape change according to its diameter:

$$c_D = \begin{cases} \frac{16}{Re_b} & Re_b \leq 0.49, \\ \frac{20.68}{Re_b^{0.643}} & 0.49 \leq Re_b \leq 100, \\ \frac{6.3}{Re_b^{0.385}} & 100 \leq Re_b \leq \frac{2065.1}{We^{2.6}}, We \leq 8, \\ \frac{3}{3} & Re_b \geq 100, Re_b > \frac{2065.1}{We^{2.6}}, We \leq 8 \\ \frac{We}{3} & Re_b > 100, We > 8 \end{cases} \tag{16}$$

$W_e$ is the Weber number and defined as the ratio of inertia force and surface tension:

$$W_e = \frac{\rho d_b |\mathbf{u} - \mathbf{u}_b|^2}{\sigma_b} \tag{17}$$

where $\sigma_b$ is the surface tension coefficient of bubble. The virtual mass force is an additional force produced in the process of the bubble being accelerated relative to liquid phase. The virtual mass coefficient is equal to 0.5, which is the theoretical value of a spherical particle moving in fluid:

$$\mathbf{F}_{VM} = C_{VM} \frac{\rho}{\rho_b} (\mathbf{u}_b \nabla \mathbf{u} - \frac{d\mathbf{u}_b}{dt}), \tag{18}$$

where $C_{VM}$ is virtual mass force coefficient. The pressure gradient force is an additional force arising from the pressure gradient in the fluid and always pointing from the region of high pressure to low pressure:

$$\mathbf{F}_P = \frac{\rho}{\rho_b} \mathbf{u}_b \nabla \mathbf{u} \tag{19}$$

When the bubble rises in the fluid, there exists an unsymmetrical pressure distribution on its external boundary. Pressure is the lowest in the region of the largest relative velocity, and, therefore, the bubble is driven into this region due to a lift force. The lift force depends on the vector product of the slip velocity and the curl of the liquid velocity, and, therefore, the lift force acts in a direction perpendicular to both the slip velocity and the curl of the liquid velocity field. Regarding to the lift coefficient, the experiment value of 0.5 was used in the model:

$$\mathbf{F}_L = C_L \frac{\rho}{\rho_b} (\mathbf{u} - \mathbf{u}) \times (\nabla \times \mathbf{u}), \tag{20}$$

where $C_L$ is the lift coefficient.

### 2.2.2. Numerical Procedure

The work is conducted by adopting the finite volume method based software ANSYS Fluent. The computation is conducted by a transient pressure-based solver with a chosen time step 0.01 s. The SIMPLEC scheme is adopted for pressure–velocity coupling. The Geo-Reconstruct scheme is used for solving volume fraction equation. The pressure staggering option scheme (PRESTO!) is selected for the pressure equation. The convergence criteria are set to $10^{-5}$ for the residuals of the continuity equation, the momentum equation, and the transport equations of $k$ and $\epsilon$.

### 2.2.3. Boundary Conditions, Mesh Sensitivity Test, and Validation

Figure 2a shows the boundary conditions applied. The Argon flow rate is applied on the plug surfaces. The bubbles are assumed to dissolve in the air region after the bubbles break through the slag layer and they are eliminated from a calculation domain.

A mesh sensitivity test has been carried out by using four different meshes for which the corresponding total element numbers are 259,148 (Mesh 1), 313,700 (Mesh 2), 417,952 (Mesh 3), and 477,330 (Mesh 4), respectively. Table 2 showed the details of the main characteristics of the different meshes and errors in different variables.

It is found that the meshes with Mesh 3 and Mesh 4 produced quite similar simulation results. Thereafter, all the simulations are based on mesh Mesh 3, which ensures a good precision at a reasonable computational cost. Figure 2b shows the details of Mesh 3.

Figure 2c showed a cross-validation. We use our result (liquid axial velocity) to compare the water model experimental result obtained by Sheng and Irons [20]. They have good agreement.

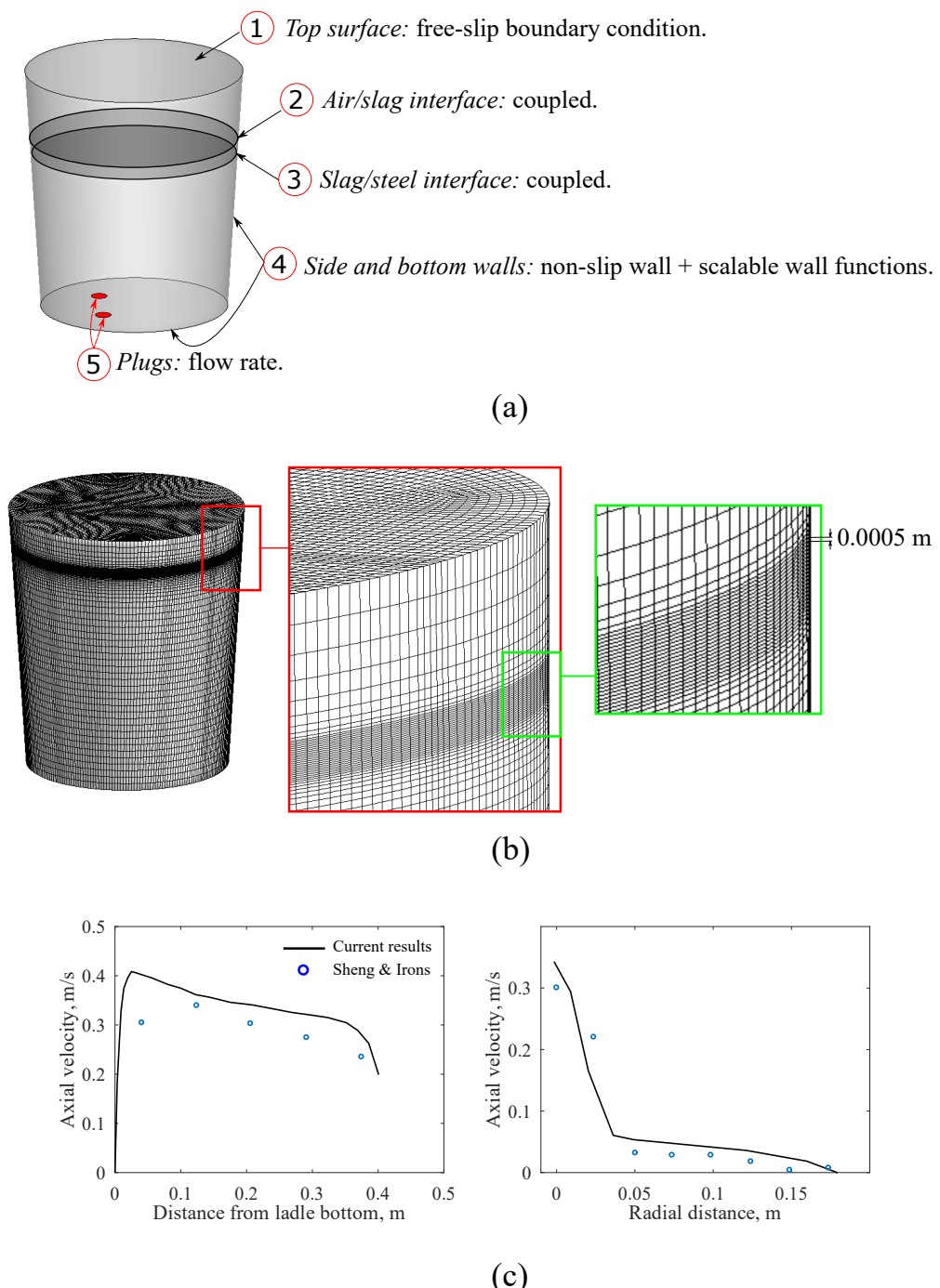

**Figure 2.** The adopted boundary conditions (**a**), the mesh details (**b**), and the cross-validation (**c**).

**Table 2.** Main characteristics of the different meshes and errors in averaged velocity, turbulence kinetic energy, and the dissipation rate of turbulent kinetic energy, respectively, on the central section of the domain, to the reference mesh, Mesh 4. The Argon flow rate is 120 NL/min.

|  | Mesh 1 | Mesh 2 | Mesh 3 | Mesh 4 |
|---|---|---|---|---|
| total element number | 259,148 | 313,700 | 417,952 | 477,330 |
| $E_{|\mathbf{u}|}$ | $4.23 \times 10^{-2}$ | $3.22 \times 10^{-2}$ | $2.91 \times 10^{-3}$ | - |
| $E_{\bar{k}}$ | $1.20 \times 10^{-2}$ | $3.96 \times 10^{-3}$ | $3.01 \times 10^{-4}$ | - |
| $E_{\bar{\epsilon}}$ | $7.09 \times 10^{-2}$ | $4.62 \times 10^{-2}$ | $4.66 \times 10^{-4}$ | - |

### 2.2.4. Current Simulation Strategy

To answer the questions raised in Section 1, we conducted 16 numerical simulations, namely Cases 1 to 16, respectively, as Table 3 shows.

**Table 3.** The different conditions for the conducted cases in the current work. The dimensionless flow rate per plug $Q^*$ is normalized by the $g^{0.5}h_{bh}^{2.5}$, where $b_{bh}$ is the bath height.

|  | No. of Plug (-) | Injected Bubble Distribution (-) | Flow Rate per Plug (NL/min) | $Q^*$ (-) | Bubble Diameter (m) |
|---|---|---|---|---|---|
| Case 1 | 1 | Uniform | 120 | $4.098 \times 10^{-5}$ | 0.006298 |
| Case 2 | 1 | Uniform | 170 | $5.806 \times 10^{-5}$ | 0.006833 |
| Case 3 | 1 | Uniform | 300 | $1.025 \times 10^{-4}$ | 0.008033 |
| Case 4 | 1 | Uniform | 600 | $2.049 \times 10^{-4}$ | 0.009773 |
| Case 5 | 1 | Logarithm-normal function | 120 | $4.098 \times 10^{-5}$ | - |
| Case 6 | 1 | Logarithm-normal function | 170 | $5.806 \times 10^{-5}$ | - |
| Case 7 | 1 | Logarithm-normal function | 300 | $1.025 \times 10^{-4}$ | - |
| Case 8 | 1 | Logarithm-normal function | 600 | $2.049 \times 10^{-4}$ | - |
| Case 9 | 2 | Uniform | 60 | $2.049 \times 10^{-5}$ | 0.005543 |
| Case 10 | 2 | Uniform | 85 | $2.903 \times 10^{-5}$ | 0.005814 |
| Case 11 | 2 | Uniform | 150 | $5.123 \times 10^{-5}$ | 0.006662 |
| Case 12 | 2 | Uniform | 300 | $1.025 \times 10^{-4}$ | 0.008033 |
| Case 13 | 2 | Logarithm-normal function | 60 | $2.049 \times 10^{-5}$ | - |
| Case 14 | 2 | Logarithm-normal function | 85 | $2.903 \times 10^{-5}$ | - |
| Case 15 | 2 | Logarithm-normal function | 150 | $5.123 \times 10^{-5}$ | - |
| Case 16 | 2 | Logarithm-normal function | 300 | $1.025 \times 10^{-4}$ | - |

The maximum argon flow rate selected is 600 NL/min, and it is the maximum value for a gas to inject into the ladle through a porous plug [21]. For Cases 1–4 and Cases 9–12, the injected bubble size is uniform. The bubble diameter ($d_b$) is obtained by the empirical equation [22]:

$$d_b = [(\frac{6\sigma d_0}{\rho_l g})^2 + 0.0242(Q_g^2 d_0)^{0.867}]^{1/6},\tag{21}$$

where $\sigma$, $d_0$, $\rho_l$, $g$, and $Q_g$ denote the surface tension between the gas and the liquid steel, the gas inlet diameter, density of the liquid steel, gravity, and the flow rate of the gas, respectively. For Cases 5–8 and Cases 13–16, the Logarithm-normal function distribution cases, the bubble distribution ($P$) is given by [15]:

$$P = P_m exp\{-[ln(d_b) - ln(d_b^m)]^2 / [ln(s)]^2\},\tag{22}$$

where $P_m$ denotes the maximum relative probability and $s = 0.026$ m, representing the standard deviation of the distribution. $d_b^m$ is is defined as the diameter of the bubble which has the maximum number distribution:

$$d_b^m = 0.04(\frac{Q_g^2}{g}) + 0.0007,\tag{23}$$

Equation (22) is implemented through a user defined function (UDF) to the finite volume codes.

### 3. Overview of the Injection Process

Figure 3 shows an overview of the argon bubble injecting into the ladle with the conditions of Case 5.

There are key stages in the whole process. These consist of the bubble being injected from the plug (Figure 3a: $t_s = 0.5$ s), the bubbles traveling in the bulk melt (Figure 3b: $t_s = 2.5$ s), the leading bubble reaching the slag layer bottom surface (Figure 3c: $t_s = 5$ s), the bubbles traveling in the slag layer (Figure 3d: $t_s = 5.5$ s), the leading bubble reaching the slag layer top surface and generating the slag eye (Figure 3e,f: $t_s = 6$–9 s), and the process being stabilized (Figure 3g,h: $t_s = 35$–45 s), respectively.

Clearly, the time scale for the Stages from I to V is negligible compared to the ladle refining process: 45 min for a 210-tonne ladle [1]. Therefore, in the following comparison, we only compare the results in the stage whilst the flow is stabilized.

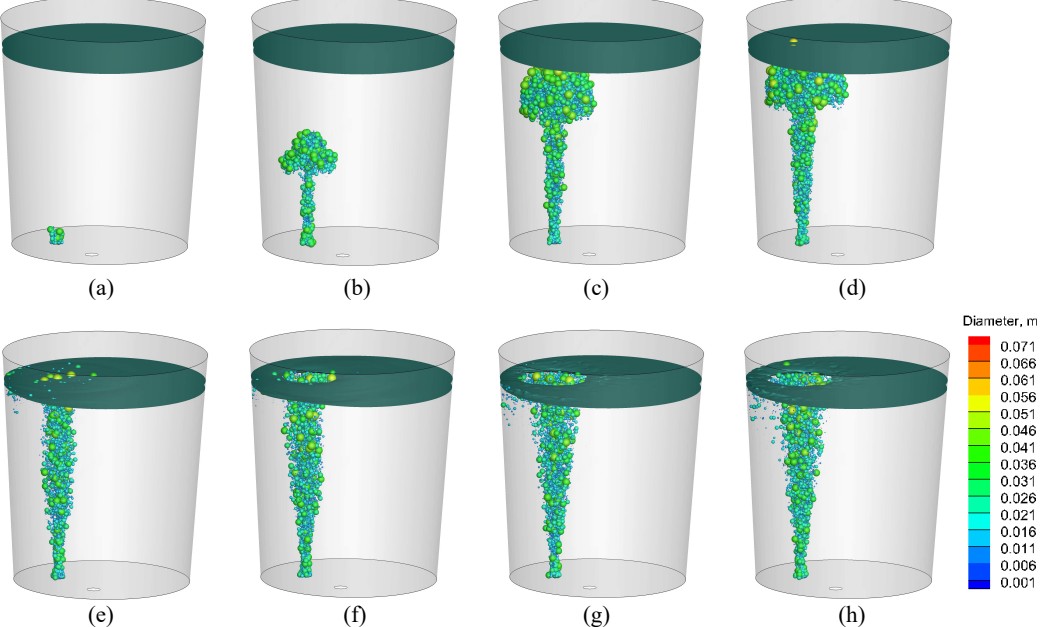

**Figure 3.** Overview of the argon bubble injection to the ladle (Case 5): (**a**) Stage I: bubble injected from plug ($t_s = 0.5$ s); (**b**) Stage II: bubbles travel in the bulk melt ($t_s = 2.5$ s); (**c**) Stage III: leading bubble reaches the slag layer bottom surface ($t_s = 5$ s); (**d**) Stage IV: bubbles travel in the slag layer ($t_s = 5.5$ s); (**e**,**f**) Stage V: leading bubble reaches the slag layer top surface and generates the slag eye ($t_s = 6$–9 s) and (**g**,**h**) Stage VI: the process is stabilized ($t_s = 35$–45 s).

## 4. Bubble Distribution during the Process

Figure 4 shows the weight ratio of the bubbles of different bubble diameters for Cases 1 to 16.

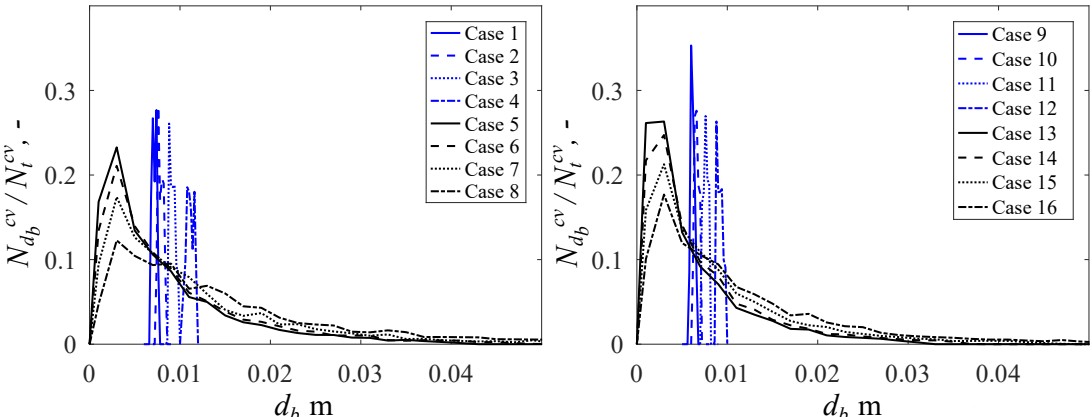

**Figure 4.** The weight ratio of the bubble vs. bubble diameter for Cases 1 to 16. Injected bubble size distribution for both Log-normal function (black curves) and uniform (blue curves). $N_{d_b}^{cv}$ and $N_t^{cv}$ denote the number of the bubble for a given diameter ($d_b$) and the total number of the bubble in a controlled volume.

The weight ratio is defined as $N_{d_b}^{cv}/N_t^{cv}$. $N_{d_b}^{cv}$ and $N_t^{cv}$ denote the number of the bubble for a given diameter ($d_b$) and the total number of the bubbles in a controlled volume. In the figure, the controlled volume height is 1.1 to 2.1 m from the ladle bottom. For Figure 4 (Left), the results clearly showed the differences between Cases 1–4 (blue, uniform) and Cases 5–8 (black, logarithm-normal function): Cases 1–4 had the most popular bubble diameter in the range of the injected bubble diameter values: for example, for Case 1, the injected bubble diameter is 0.006298 m and the bubble diameter range is from 0.0068 to 0.0076 m after the process is stabilized. The diameter difference between the injected value and the final value is due to the bubbles growing as they travel from the bottom to the top region of the ladle. The main reason for the diameter variation is due to the pressure difference at different heights of the ladle: any local pressure changes can result in the change of the bubble shape, which is discussed well in the previous work [23]. There is no bubble with a diameter above 0.0076 m that was found in the given volume. However, for Cases 5–8, the bubbles with different diameters obey the Log-normal distribution. The maximum bubble diameter is 0.041 m ($N_{d_b}^{cv}/N_t^{cv}$ = 0.0013). The number of the porous plugs will not change this trend, as showed in Figure 4 (Right). Figure 5 shows the bubble distribution for different cases at a simulation time of 40 s.

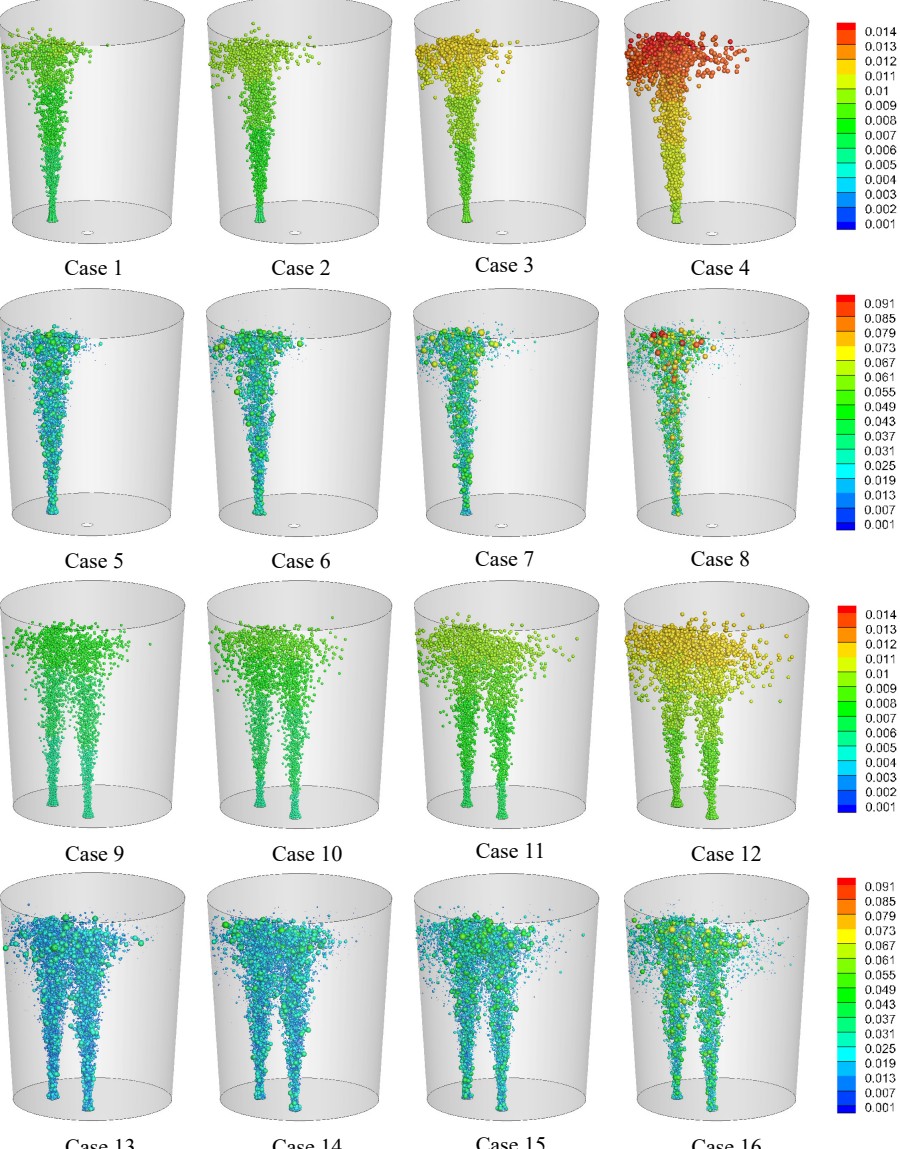

**Figure 5.** Argon bubble distribution for different cases at $t_s$ = 40 s. The legend represents the bubble diameter $d_b$.

The results indicated that, for Cases 1–4 and 9–12, the bubble diameter increases as the bubble travels and departs from the ladle bottom. For a given height, a relative uniform diameter distribution is found. This is due to in the simulation, the bubble interactions, e.g., the breaking up and the coalescence, are not considered. For Cases 5–8 and 13–16, the bubble grows as it travels; however, this is different compared to Cases 1–4 and 9–12, for a given height, for which a mixed size of bubbles can be found. This is related to the bubble distribution at the porous plug. The Log-normal function distribution was proved to be closer to the real case, both numerically [16] and experimentally [15]. The mechanics of these differences affecting the stirring process, e.g., the slag eye, are the issues that we are going to discuss in the next section.

## 5. Slag Eye Comparison

Figure 6 showed the snapshots of the slag eye at $t_s$ = 40 s for Cases 1 to 16.

The results indicated that, as the flow rate is increasing, the turbulence kinetic energy ($k$) is increased as well and more turbulence is presented. It is also observed that the area of the slag eye increases as the argon flow rate is increased. The dual plugs cases (Cases 9–16) have a larger slag eye compared to the single plugs cases (Cases 1 to 8) under the same total flow rate.

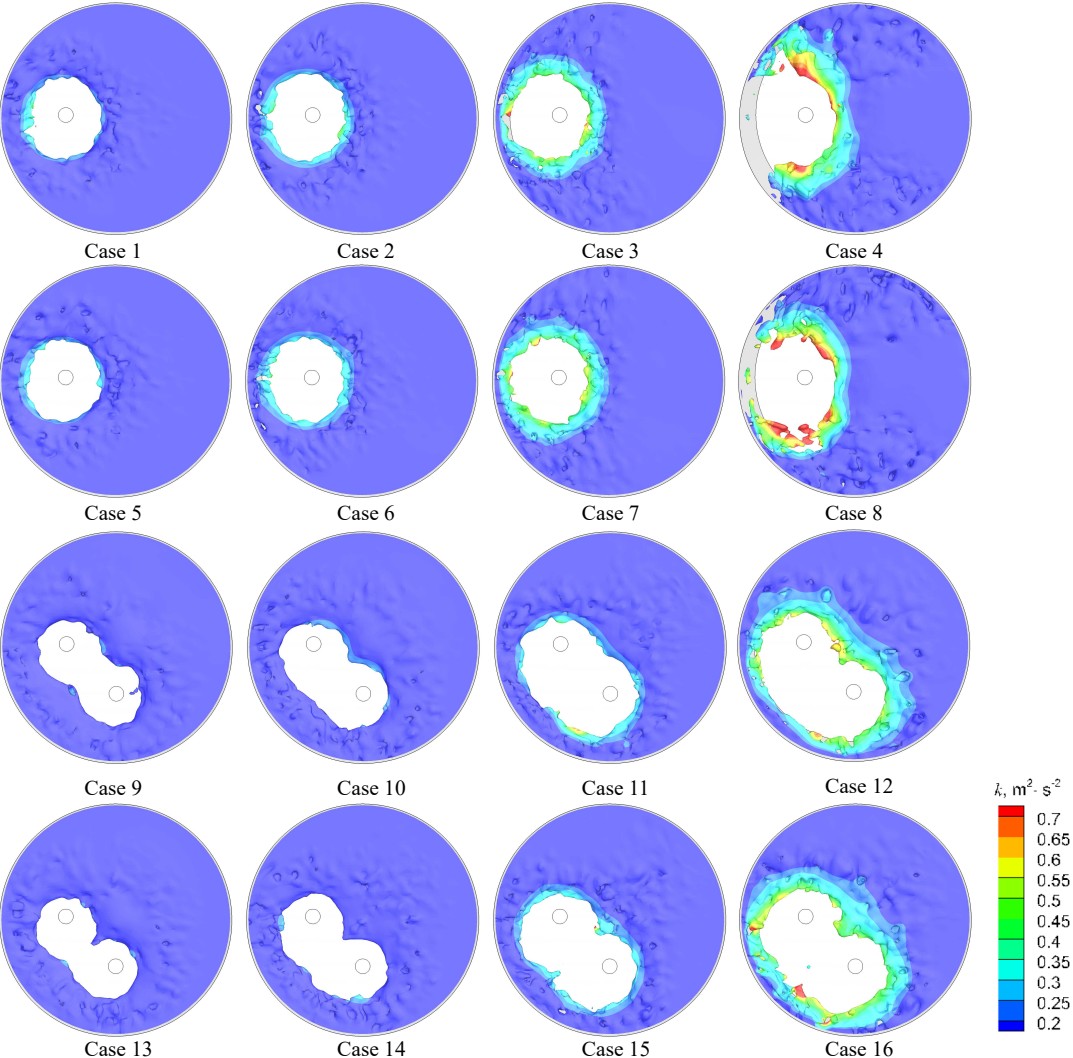

**Figure 6.** The slay eye snapshots for Cases 1 to 16 at $t_s$ = 40 s.

Figure 7 shows the relationship between dimensionless slag eye $A^*$ and dimensionless flow rate $Q^*$.

$A^*$ is normalized by the total area of the slag surface. The dimensionless flow rate per plug is normalized by the $g^{0.5}h_{bh}^{2.5}$, where $b_{bh}$ is the bath height [24]. The error bar showed the area values at different $t_s$. The results showed that the slag open eye area increases as the gas flow rate is increasing. This trend is independent from the porous plug numbers. It is also observed that the uniform bubble distribution cases (Cases 1–4 and Cases 9–12, in black) predict larger slag eye area compared to the the Log-normal function bubble size distribution cases (Cases 6–8 and Cases 13–16, in red). The difference becomes more obvious at a higher Argon flow rate, e.g., Cases 4 and 8, with an area increasing 15.8%. This trend is true for both a single plug case and dual plugs cases, as showed in the figure. This difference is worthy of further investigation and understanding the reason for this will benefit the understanding of the question raised in Section 1. We now only focus on the high Argon flow rate cases: Cases 4 and 8.

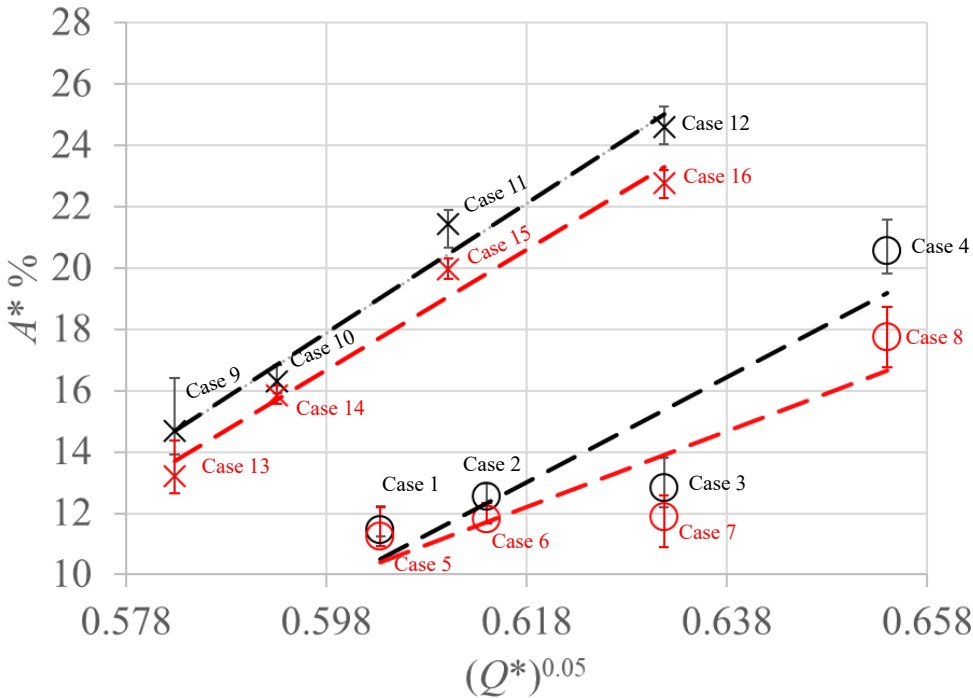

**Figure 7.** The dimensionless open eye area ($A^*$ %) vs. the dimensionless argon flow rate per plug ($Q^*$). $A^*$ is defined as the ratio of the open eye area to total area of the slag surface. Black and red represent the cases with uniform and logarithm bubble diameter distribution, respectively.

Figure 8 shows the comparisons between the axial velocity ($Top$) and the turbulent kinetic energy ($Bottom$) for Cases 4 and 8.

In the figure, $h$ is defined as the distance from the measured point to the ladle bottom plane. $H$ is the height of the ladle. The red dashed line ($h/H = 0.857$) represents the slag/steel interface. The axial velocity and the $k$ are captured along a line from ladle bottom to top across the center of the porous plug. The results, as showed in Figure 8 Left, indicated that the maximum axial velocity appears at the location close to the ladle bottom region, e.g., $h/H = 0.2$. This agrees with the results obtained by Mazumdar et al. [25]: $h/H = 0.25$. This is because the bubbles were injected at a higher velocity. However, this momentum is dissipated within a very short distance of the injector [26]. This phenomenon is independent from the gas flow rate up to 600 NL/min, according to the current study. It is also observed that the Log-normal function bubble diameter case (Case 8) has a higher

magnitude of velocity compared to the uniform function bubble diameter case (Case 4). This difference is small at a small gas flow rate, e.g., 120 NL/min (Case 1), which agrees with the work done by Chen & He [27]. However, this difference becomes obvious when the flow rate is increased. The reason for this is due to the size distribution of the bubble being different. In the range of the bubble just injected, e.g., $h/H = 0$ to 0.15, there is little velocity difference. This is because, in this stage, the velocity is dominant by the momentum injected from the plug. Cases 4 and 8 have the same flow rate (600 NL/min). After that, the plume is driven by the buoyancy of the rising bubble and the difference is presented. This difference is also reflected by the turbulence kinetic energy, as showed in Figure 8 Right. A lager $k$ value is presented for Case 8 compared to Case 4. Figure 9 shows the snapshots of the contour of the turbulent kinetic energy for Cases 4 (Left) and 8 (Right) at $t_s = 40$ s.

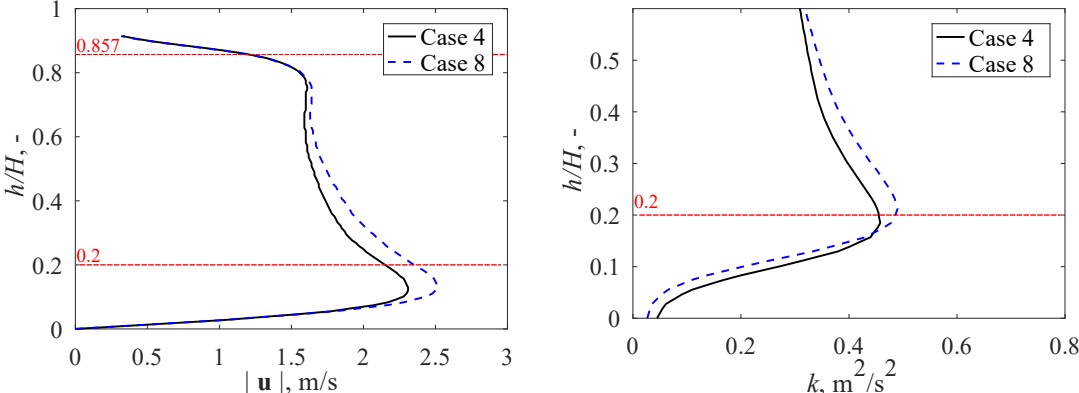

**Figure 8.** (**Left**): Axial velocity comparison for Cases 4 and 8. The velocity is captured along the line vertical to the ladle bottom and through the center of the porous plug. $h$ is defined as the distance from the measured point to the ladle bottom plane. $H$ is the height of the ladle. The red dashed line ($h/H = 0.857$) represents the slag/steel interface. (**Right**): turbulent kinetic energy comparison between Case 4 and 8. The data are averaged between 35–45 s.

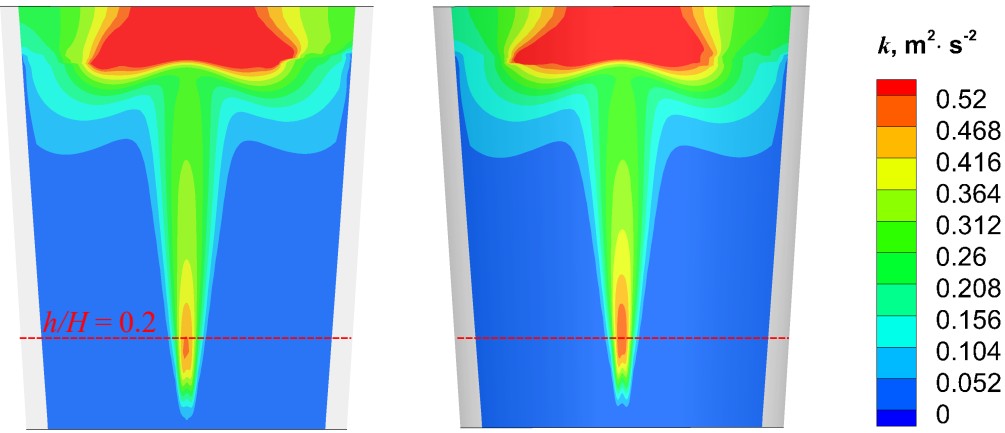

**Figure 9.** The snapshots of the contour of the turbulent kinetic energy for Cases 4 (**Left**) and 8 (**Right**) at $t_s = 40$ s.

The contours are plotted on the cross-section ($y - z$ plane) where $x = -0.665$ m. The center of the plug is located on this plane. These results showed that a higher turbulence level was found for the Log-normal function case (Case 8, Figure 9 Right) compared to the uniform case (Case 4, Figure 9 Left).

Figure 10 showed the snapshots of regions of slag eye with bubbles and the magnitude of plume velocity at $t_s = 40$ s. (a) and (c): Case 4; (b) and (d): Case 8.

The results showed, for Case 4 Figure 10a, more bubbles with a uniform diameter present in the vicinity of the slag eye area and also moving to the the ladle wall region; this movement may help enlarge the eye area. In contrast, for Case 8, the bubbles with larger diameters are more concentrated on the eye region, as showed in Figure 10b. Figure 10c,d shows the magnitudes of the plume velocity. The maximum values are between 0.8 to 1 m/s. The velocity values fit the empirical plume velocity equation perfectly: $u^{max} = 8.64 \, Q^{0.25}$ [19].

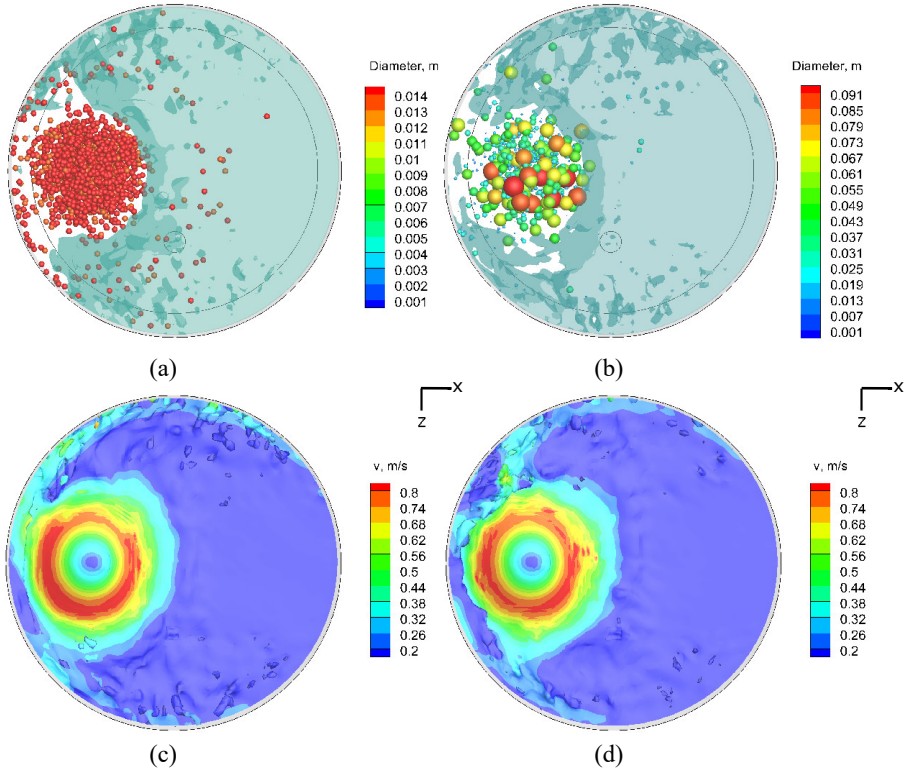

**Figure 10.** The snapshots of regions of slag eye with bubbles and the magnitude of plume velocity at $t_s = 40$ s. (**a**,**c**): Case 4; (**b**,**d**): Case 8.

## 6. Conclusions

This work investigated the influence of the injected/initial bubble size distribution on main characteristics in the gas stirring ladle process, such as slag open eye. Two distributions were compared: uniform and the Log-normal function. The bubble size of uniform distribution was calculated through the empirical equation in the work of Mori et al. [22]. The Log-normal function distribution of bubble size was implemented into a program based on the finite volume method through a user defined function. The influence of the porous plug number and the gas flow rate were also investigated. Main conclusions can be summarized as follows:

1.  Without considering the bubble interactions, the bubble size distribution remains in its initial injected distribution.

    For both distributions, the bubble size increases as it floats up. For the cases with uniform distribution, the bubbles in the region at a given height have similar sizes.

2.  At a low gas flow rate, e.g., ≤300 NL/min, for two bubble size distributions, the flow behavior in the ladle (bulk melt zone) has little difference. At a high gas flow rate, e.g., 600 NL/min, a relative higher axial velocity is obtained from a Log-normal function distribution compared to the uniform distribution at the location near the ladle bottom, e.g., $h/H = 0.2$. This difference is because the different bubble size clusters with different bubble sizes play a more dominant role in stirring the liquid steel. Both cases with single plug and dual plugs follow this trend.

3.  In terms of the area of the slag open eye, the results indicated that a larger eye area is predicted by the uniform bubble size distribution cases compared to the Log-normal function distribution. The increasing of the flow rate and the number of the plug number make this phenomenon more evident, e.g., 15.8% area increase at the gas flow rate equals 600 NL/min. This indicated to us that this eye area difference should be taken into account if we use uniform bubble distribution, without taking bubble interaction into account, with an aim to predict the real situation.
4.  In general, uniform bubble size distribution could predict the flow and slag eye features with low to medium gas flow rates. However, at a high gas flow rate, e.g., 600 NL/min, the results' differences need to be noticed, e.g., under-predicting the axial velocity and over-predicting the slag eye.

Future work will focus on the simulations and experiments with the aim to investigate the ladle vibrations induced by gas stirring in order to figure out better methods to measure and control the ladle vibrations.

**Author Contributions:** The authors contributed equally to this work. All authors have read and agreed to the published version of the manuscript.

**Funding:** This research was funded by the Major Technology Project of China National Machinery Industry Corporation (SINOMACH): "Research and application of key technologies for industrial environment monitoring, early warning and intelligent vibration control (SINOMAST-ZDZX-2017-05 )" and the APC was funded by the above-mentioned Project (SINOMAST-ZDZX-2017-05 )".

**Acknowledgments:** The authors would like to thank the reviewers for their work that has contributed to this paper.

**Conflicts of Interest:** The authors declare no conflict of interest.

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
