# Peer review of "Influence of the Gas Bubble Size Distribution on the Ladle Stirring Process"

_processes, doi:10.3390/pr8121663_

Round 1

Reviewer 1 Report

Article presents the analysis of the bubble size distribution in the ladle stirring process of molten iron by means of the CFD modelling. Authors focused mainly on the impact of bubbles on the slag eye size. The language through the Manuscript is correct and methodology is clear. This article is publishable after some minor revision. The reviewer suggests to correct some points that will improve the quality of the work.

General remark: The manuscript is 16 pages long, but the manuscript pdf file size is 53,3 MB. The case of this is probably high resolution of the used images, which is usually a good thing. Obviously, I would not insist in changing the picture size. However, Authors should consider decreasing of the overall file size to make it more reader-friendly (if possible).

1) Page 4, line 84:where σk, σε, C1 and C2 have constant values 1.0, 1.3, 1.38 and 1.92, respectively” – all the constant values were set to their default for standard k-ε model, except C1 (default value 1,44). Why Authors changed the value of C1?

2) Page 7, Table 3 – the gas flow rate shown in the table is a total flow rate or flow rate per single plug?

3) Page 7, Table 3 – I think, that Cases 1-4 and 9-12 should not be directly compared with corresponding Cases 5-8 and 13-16. In my opinion, bubble size distributions should have the same value of the median bubble size (or mean bubble size) for corresponding Cases. Please comment.

4) Page 7, lines 139-140: “(…) Qg denote (…) the flow rate of the gas (…)” – but in the equation (23) Authors use the symbol Q for gas flow rate. Symbols used in all of the equations should be consistent.

5) Page 8, lines 143-144: “(…) s denote (…) the standard deviation of the distribution” – what was the values of the standard deviation used in this study?

6) Page 8, line 144: “dbm is the maximum bubble size” – according to log-normal distribution model, dbm should stand for median bubble size, not maximal bubble size.

7) Page 12, line 190: “A* is normalized by the total area of the slag surface” – If I understood it correctly, A* is the ratio of the slag eye area to the cross section area of the apparatus at the height of slag surface? If so, the value of A* should vary from 0 to 1, but in the Fig. 7 its value varies from about 10 to 25. Why?

Reviewer 2 Report

The paper with Title of Influence of the gas bubble size distribution on the ladle stirring process written by Mengkun Li andLintao Zhang is about the figuring out the influence of gas bubble size distribution on the ladle stirring process. The work is conducted through three-dimensional numerical simulation based on the finite volume method. The authors investigate two distributions, uniform and Log-normal function,  under different gas flow rates and number of porous plugs. By the way is what clear that the axial velocity and the area of the slag eye, have difference for high flow rate. The authors should use non dimensional sizes to generalize their study. As well it is not clear how the Log-normal function is a better choice to predict the melt behaviour and the slag open eye while the other functions not tested.
